# Total Knee Replacement prediction using Structural MRIs and 3D Convolutional Neural Networks

**Tianyu Wang**[1]                                         TW1682@NYU.EDU
**Kevin Leung**[2]                                          KL2596@NYU.EDU
**Kyunghyun Cho**[1]                               KYUNGHYUN.CHO@NYU.EDU
**Gregory Chang**[3]                     GREGORY.CHANG@NYULANGONE.ORG
**Cem M. Deniz**[3]                          CEM.DENIZ@NYULANGONE.ORG

[1] *Center for Data Science, New York University*

[2] *Leonard N. Stern School of Business, New York University*

[3] *Department of Radiology, New York University School of Medicine*

**Editors:** Under Review for MIDL 2019

## Abstract

Osteoarthritis (OA) is a chronic degenerative disorder of joints and is the most common reason leading to total knee joint replacement (TKR). In this work, we developed an automated OA-relevant imaging biomarker identification system based on MR images and deep learning (DL) methods to predict knee OA progression. Our results indicate that the combination of multiple MR images with different contrast and resolution provides the best model to predict TKR with AUC 0.88±0.02.

**Keywords:** MRI, Image Classification, Convolutional Neural Networks, Osteoarthritis, Total Knee Replacement

## 1. Introduction

OA is the most common form of arthritis and the major cause of physical disability in the elderly. The discovery of imaging-based biomarkers could lead to the identification of new treatment targets and mechanisms for shorter, more efficient clinical trials of possible disease-modifying agents. In recent years, MRI has been increasingly used in OA, since it can visualize all tissues in the knee joint involved in OA, such as cartilage, menisci , bone and soft tissue (Carballido-Gamio et al., 2011; Kazakia et al., 2013). Even though knee OA is a disease of the whole joint affecting all the joint tissues, current image analysis methods focus mainly on locations or features (Neogi et al., 2013; Eckstein et al., 2013) previously defined to characterize knee joint health separately.

In this study, we used DL methods to identify OA-relevant imaging biomarkers directly from the whole knee joint using knee MR images without focusing on pre-defined locations and features. Our goal is to predict the outcome of OA as the subjects probability of TKR within 9 years directly from MR images acquired using different contrast and resolution. Our convolutional-based approach significantly improves the outcome prediction model based on clinical risk factors by providing image processing insights on knee OA from a DL perspective.

## 2. Outcome Prediction Model

We developed two 3D CNN models (DESS and TSE models) using residual blocks (He et al., 2016) to predict the binary outcome variable using structural MR images. Input for these models were either DESS or TSE images providing different contrast and resolution images from the knee. First two convolutional layers and a max pooling layer were used to bring the spatial volume of the feature maps into a same range for both images. The first convolutional kernel for TSE model was of size 7x7x3 and strides 2,2,2 followed by a max-pooling layer over a 3x3x3 window with stride 2x2x1. The first convolutional kernel for DESS model was of size 7x7x7 and strides 2,2,2 followed by a max-pooling layer over a 3x3x3 window with stride 2x2x2. The second convolution kernel was with size 3x3x3 and stride 2x2x1 for both models. After reducing the spatial dimension, both models use the same 8 residual blocks (He et al., 2016) structure. The last residual block was followed by a global max pooling layer and a fully connected layer of size 256. Models were trained using a 4-fold CV and Adam optimizer with lr=2e-4.

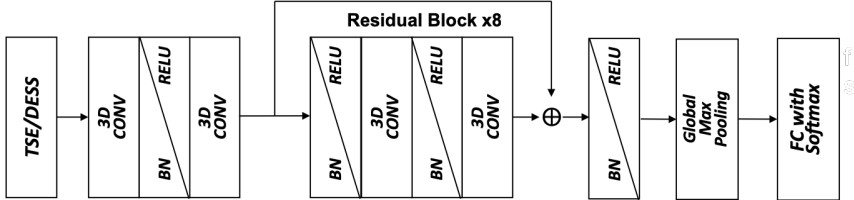

Figure 1: TSE and DESS Model Architectures

Multiple TKR outcome prediction models based on logistic regression (LR) was developed using the output of the 3D CNN models, and subjects' demographic and clinical information: baseline age, gender, race, BMI, KOOS Quality of Life score (Roos et al., 1998) and WOMAC pain score (Bellamy et al., 1988). Receiver operating curve (ROC) analysis and DeLong test on the area under the ROC (AUC) were used to assess the performance of developed outcome prediction models.

## 3. Data Set

Sagittal intermediate-weighted fat-saturated Turbo Spin Echo (TSE) images and sagittal 3D double-echo steady-state (DESS) with water excitation images from Osteoarthritis Initiative (Peterfy et al., 2008) were used. In general, TSE images provides information about cartilage loss, ligament integrity, meniscal tears and subchondral bone marrow. DESS images provide information about total joint cartilage morphology, bone area and shape, and osteophytes. The spatial resolution $(mm^3)$ / matrix size for the images were 0.357x0.511x3.0 / 448x448x37 and 0.365x0.365x0.7 / 384x384x160 for TSE and DESS images, respectively.

718 case-control pairs (age: 63±12 years, BMI: 29±6.6 kg/m , 274/444 male/female split) were selected from OAI dataset by propensity score matching on individuals based on the baseline variables: age, BMI, gender and race. Cases were defined as individuals who received a medically confirmed TKR after baseline. We defined controls as individuals who did not receive a TKR in either knee on the 108-month visit.

## 4. Results

Both TSE and DESS models achieved the AUC of 0.86±0.01 that is significantly higher than the baseline LR model using demographic and clinical information only (LR-clinical) (AUC: 0.77±0.02, p<0.01). Incorporating the output of both DL models into baseline LR-clinical model further improved the prediction performance (AUC: 0.88±0.02, p<0.01).

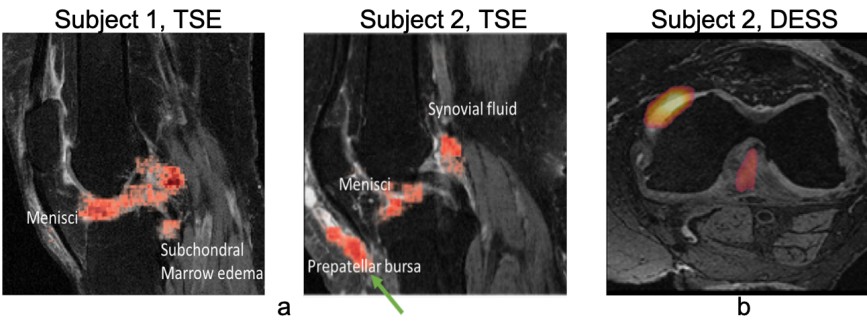

Figure 2: Visualization of the regions in which TSE-model and DESS-model use to make a decision with high impact. (a) the occlusion heatmap under TSE model overlaid on sagittal MRI view from two subjects. (b) the heatmap generated by DESS model overlaid on an axial MRI view of Subject 2.

Occlusion maps (Zeiler and Fergus, 2014) were used to understand the CNNs learning behavior by analyzing discriminative image regions used by a CNN to identify the subjects TKR outcome. Figure 2a shows that the TSE-model model pays attention to the locations on which where OA has an immediate impact, such as cartilage, as the probability of the correct class drops significantly. More interestingly, as indicated by the green arrow in the second row, TSE-model also uses the information from the front of the knee cap where prepatellar bursitis was diagnosed for this patient. Moreover, DESS-model incorporates the complementary information from cartilage, posterior intercondylar notch and posterior cruciate ligament to predict the TKR outcome for this subject (Figure 2b).

## 5. Conclusions and Future Work

We present an automated method to identify OA-relevant imaging biomarkers for the prediction of TKR probability using 3D CNN and structural MR images. Our preliminary results suggest that the use of 3D CNNs provides a significantly better estimation of TKR probability within 9 years compared to a model using demographic and clinical information. Using the output of multiple DL models trained on MR images with different contrast and resolution helped to improve the predictive power of the outcome prediction model.

In the future, we plan to design a general model to incorporate other available MR imaging sequences from OAI dataset into our CNN-based modeling approach. In addition, we would like to see how DL models perform against quantitative mean cartilage thickness calculated in the central medial tibiofemoral compartment (Eckstein et al., 2013).

## Acknowledgments

The OAI is a public-private partnership comprised of five contracts (N01-AR-2-2258; N01-AR-2-2259; N01-AR-2-2260; N01-AR-2-2261; N01-AR-2-2262) funded by the National Institutes of Health, a branch of the Department of Health and Human Services, and conducted by the OAI Study Investigators. Private funding partners include Merck Research Laboratories; Novartis Pharmaceuticals Corporation, GlaxoSmithKline; and Pfizer, Inc. Private sector funding for the OAI is managed by the Foundation for the National Institutes of Health. This manuscript was prepared using an OAI public use data set and does not necessarily reflect the opinions or views of the OAI investigators, the NIH, or the private funding partners.

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
