# OpenReview forum: "Total Knee Replacement prediction using Structural MRIs and 3D Convolutional Neural Networks"
_MIDL.io/2019/Conference/Abstract — MIDL Abstract 2019_

### Official Review · AnonReviewer2 · 2019-04-29
**Straightforward but well-executed application abstract, key details on data splits and training missing**

**Rating:** 3
**Confidence:** 3

**Review:**

The authors train two models to predict whether total knee replacement is needed for different MRI sequences. The abstract is pretty straightforward and the results are decent. The key problem with the abstract is the lack of details on data splits; what was used for training, validation and testing?

---

### Official Review · AnonReviewer1 · 2019-05-01
**Sensible application of DL for image-based prediction of knee osteoarthritis progression, suitable for an abstract.**

**Rating:** 3
**Confidence:** 2

**Review:**

The abstract investigates the use of imaging biomarkers (TSE or DESS modalities) instead of, or in addition to, demographics and clinical information to predict knee osteoarthritis progression.

The image-based models are standard 3D CNN architectures (w/ residual connections, FC/softmax final layer). In addition it explores the use of occlusion maps to identify what spatial context the algorithm is basing its decision on (interpretable DL).

The presentation is clear. While there is no focus on methodological novelty, the approach is reasonable. The results and comments are suitable for an abstract.

Maybe performance under/robustness to variations in the test data (e.g. expected variations in the preprocessing or acquisition protocol if relevant) could be assessed next.

---

### Decision · Program_Chairs · 2019-05-06
**Acceptance Decision**

Accept